# Realistic phase diagram of water from "first principles" data-driven quantum simulations

Sigbjørn Løland Bore [1] & Francesco Paesani [1,2,3,4] ✉

Since the experimental characterization of the low-pressure region of water's phase diagram in the early 1900s, scientists have been on a quest to understand the thermodynamic stability of ice polymorphs on the molecular level. In this study, we demonstrate that combining the MB-pol data-driven many-body potential for water, which was rigorously derived from "first principles" and exhibits chemical accuracy, with advanced enhanced-sampling algorithms, which correctly describe the quantum nature of molecular motion and thermodynamic equilibria, enables computer simulations of water's phase diagram with an unprecedented level of realism. Besides providing fundamental insights into how enthalpic, entropic, and nuclear quantum effects shape the free-energy landscape of water, we demonstrate that recent progress in "first principles" data-driven simulations, which rigorously encode many-body molecular interactions, has opened the door to realistic computational studies of complex molecular systems, bridging the gap between experiments and simulations.

Arguably, water is the single most important molecule on Earth, being an essential component of life[1] and being directly involved in several fundamental biological and chemical processes[2]. From a scientific standpoint, one of the most intriguing aspects of water is the contrast between its simple chemical formula and its complex behavior[3]. Liquid water exhibits several anomalous properties[4], including the well-known density maximum at 4 °C which allows fish to thrive at the bottom of icy lakes. Ordinary ice, i.e., hexagonal ice (ice $I_h$), is an extraordinary solid[5]. It has a lower density than liquid water, which makes ice float on liquid water. Ice is slippery when one walks, skates, or just stands still on it, but is sticky when one touches it[6]. Strictly speaking, ordinary ice is not even a crystalline material since it exhibits orientational disorder[7], which stabilizes the lattice structure and, consequently, raises the melting point by ~100 K compared to the melting points of other similar chemical compounds[5]. The origin of these unusual properties can be traced to the ability of the water molecules to form directional hydrogen bonds whose strength and orientation fluctuate in time and space depending on temperature and pressure[8]. As determined in the 1930s, the structure of ice follows the so-called Bernal-Fowler rules, which state that every water molecule is

hydrogen bonded to four other water molecules[9]. While constraining the spatial arrangement of water molecules to tetrahedral geometries, a vast space of energetically favorable solids exists.

The phase diagram of water keeps expanding with time through a close synergy between experiment and simulation. Pioneering measurements by Angell and co-workers[10,11] and subsequent computer simulations have led to several hypotheses about the existence of a liquid–liquid critical point at deeply supercooled temperatures[12–15], which have stimulated several experimental measurements for the past two decades[16–20]. Similarly, while experiments continue to make progress in exploring the phase diagram of water[21], with 20 different crystalline ice polymorphs[22,23] and 3 amorphous forms[24] discovered to date, computer simulations have generated a plethora of energetically viable ice polymorph candidates[25]. Despite significant advancements in computer simulations, reproducing the phase diagram of water experimentally determined by Bridgeman and Taman in the early 1900s[26,27] still remains a challenge. Current state-of-the-art simulations can only qualitatively account for the equilibria between liquid water and the different ice polymorphs[28–36]. This is symptomatic of the difficulties

[1]Department of Chemistry and Biochemistry, University of California, San Diego, La Jolla, CA 92093, USA. [2]Materials Science and Engineering, University of California San Diego, La Jolla, CA 92093, USA. [3]Halicioğlu Data Science Institute, University of California San Diego, La Jolla, CA 92093, USA. [4]San Diego Supercomputer Center, University of California San Diego, La Jolla, CA 92093, USA. ✉e-mail: fpaesani@ucsd.edu

for existing water models to correctly represent the free-energy landscape of water in regions of the phase diagram that most closely resemble conditions encountered for aqueous solutions on Earth[37].

The accuracy of a computer simulation in predicting the properties of water across the entire phase diagram depends on the ability of the model used in the simulation to accurately capture the underlying molecular interactions, as well as on the extent to which the simulation exhaustively samples the free-energy landscape of water over a wide range of thermodynamic conditions. On the one hand, the free-energy landscape of water is particularly complex. For example, the average molecular dipole moment of water increases by 30–50% moving from the gas to the condensed phases[38]. Furthermore, water molecules can form highly directional hydrogen bonds whose strength is determined by many-body effects that vary significantly depending on the local three-dimensional structural arrangement[39]. In addition, due to the light mass of the hydrogen atoms, the properties of water are modulated by nuclear quantum effects, which are responsible for several differences in the behavior of light ($H_2O$) and heavy ($D_2O$) water[40,41]. On the other hand, since some ice polymorphs are separated energetically by only 0.06 kJ mol$^{-1}$[42,43], computer simulations of water's phase diagram require highly precise determination of the associated free-energy landscape.

By construction, "first principles" (or ab initio) simulations provide the most rigorous, although still approximate, description of a molecular system by solving the corresponding Schrödinger equation "on the fly"[44]. Different "first principles" methods, however, exhibit significantly different accuracy and predictive power depending on the approximations that they rely on, ranging from the Hartree–Fock method[45–47], which scales with the fourth power of the number of basis functions (that is proportional to the system size) but neglects electron correlation, to coupled-cluster methods[48,49], such as CCSD(T), i.e., a coupled-cluster method that includes single, double, and perturbative triple excitations, which currently represents the "gold standard" for molecular interactions but scales with the seventh power of the number of basis functions[50–52]. In practice, an accuracy-cost compromise has to be made in "first principles" simulations. In this context, density functional theory (DFT)[53,54], which formally scales with the third power of the number of basis functions, remains the only viable "first principles" method for computer simulations of condensed-phase systems[55]. Besides being still computationally too expensive for a complete exploration of water's phase diagram, DFT, however, suffers from inherent limitations due to the use of approximate exchange-correlation functionals and electron densities[56–64], which manifest in both functional-driven and density-driven errors[65–69]. A recent study has shown that even the most accurate DFT models exhibit errors that are similar in magnitude to the relative differences in lattice energies of ice polymorphs[70]. These findings also imply that neural network potentials of water derived from DFT-based simulations[35,71–79], which are gaining popularity in computational molecular sciences, exhibit the same limitations of the parent DFT models. Given the shortcomings associated with DFT-based simulations, it is thus not surprising that pairwise-additive water models such as TIP4P/2005[80] and TIP4P/Ice[81], which were empirically parameterized to reproduce a subset of experimental thermodynamic data, still provide some of the most reasonable representations of the phase diagram of water[28–33,82].

The development of efficient algorithms for correlated electronic structure methods has recently enabled routine coupled-cluster calculations of interaction energies for water clusters[83,84]. This has given rise to a new class of "first principles" data-driven potentials for water[85–92] that rigorously decompose the interaction energy of an arbitrary water system into individual many-body contributions[93], which can be efficiently calculated at the coupled-cluster level of theory. Among these "first principles" data-driven many-body potentials, MB-pol[89–91] exploits the "nearsightedness of electronic matter"[94] to accurately describe CCSD(T) interaction energies through a combination of machine-learned representations of short-range quantum-mechanical interactions and mean-field representations of many-body effects[95,96]. Fully derived from CCSD(T) data, MB-pol accurately predicts structural, thermodynamic, dynamical, and spectroscopic properties of water from gas-phase clusters[97–99] to the liquid[100–105] and ice[106–109] phases, bypassing the accuracy limitations of DFT-based models. The MB-pol potential is thus uniquely positioned to provide realistic, molecular-level insights into the phase diagram of water.

For a precise determination of the phase diagram of water, equally important to the accurate representation of the underlying molecular interactions is the exhaustive sampling of the associated free-energy landscape[30]. The most common approach to characterizing coexistence equilibria is thermodynamic integration, which allows for calculating free-energy differences by performing a series of simulations that connect a phase of known free energy to the phase of interest[110]. In a seminal work[28], Sanz et al. used thermodynamic integration in combination with the Einstein Molecule method and Gibbs-Duhem integration to calculate the phase diagram of water using different empirical, pairwise-additive models, providing an important benchmark for the ability of computer simulations to reproduce the experimental phase diagram. It should, however, be noted that calculating the phase diagram of water using the Einstein Molecule method is not devoid of challenges. In particular, for proton-disordered ice polymorphs, the Einstein Molecule method requires exact knowledge of the molecular configuration that minimizes the associated free energy as determined by the water model used in the simulations. This is a daunting task to accomplish for partially-disordered ice phases, such as ice III and ice V, because determining the corresponding minimum free-energy configuration requires extremely long simulations due to the extremely slow transition from one configuration to another[111,112]. Recent simulation studies carried out with the TIP4P/2005 and TIP4P/Ice force fields have shown that the Einstein Molecule method largely underestimates the thermodynamic stability of ice III compared to direct-coexistence and enhanced-coexistence simulations[33,112,113]. As discussed in the original references[33,112,113], since both direct-coexistence and enhanced-coexistence simulations explicitly simulate the crystallization process, they do not rely on any approximation for the entropic contributions associated with proton disorder. This allows for correctly determining the free-energy difference between liquid water and a given ice polymorph, independently of the extent of proton disorder present in the ice polymorph (i.e., direct-coexistence and enhanced-coexistence simulations inherently sample the relevant regions of the underlying multidimensional free-energy landscape.)

In this study, we report the phase diagram of water calculated at the fully quantum-mechanical level using the "first principles" MB-pol data-driven many-body potential. Using a multi-stage approach that leverages the computational efficiency of a deep neural network potential trained on MB-pol data (DNN@MB-pol) and rigorous free-energy sampling techniques (see Methods for details), we demonstrate that MB-pol reproduces the phase diagram of water with an unprecedented level of realism, thus closing the gap between experimental measurements and simulation predictions.

## Results
### Liquid-ice coexistence
While it is, in principle, possible to determine each coexistence line from a single melting point of the relevant ice polymorph, we calculated a total of 15 melting points. This allowed us to average the coexistence lines obtained from all the melting points, resulting in more accurate estimates. Additionally, it allowed us to compare the melting lines calculated using Gibbs-Duhem integration to the melting points directly determined from enhanced-coexistence simulations using the DNN@MB-pol potential. Figure 1a shows that the melting points determined from the enhanced-coexistence simulations lie

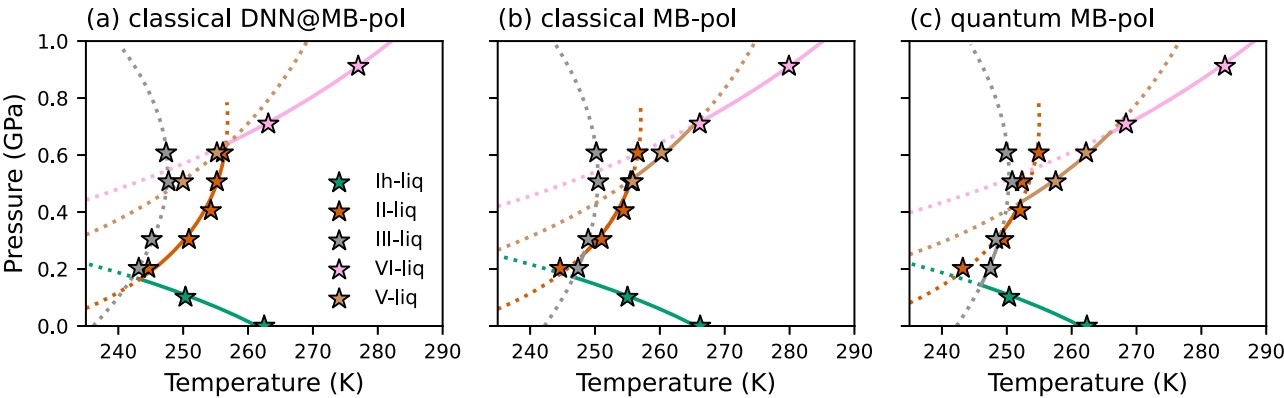

**Fig. 1 | Liquid-ice melting points and coexistence lines.** Melting points calculated at the classical level with DNN@MB-pol (**a**) and MB-pol (**b**), and at the quantum level with MB-pol (**c**) are indicated by stars, while thermodynamically stable and metastable line segments are represented by solid and dotted lines, respectively.

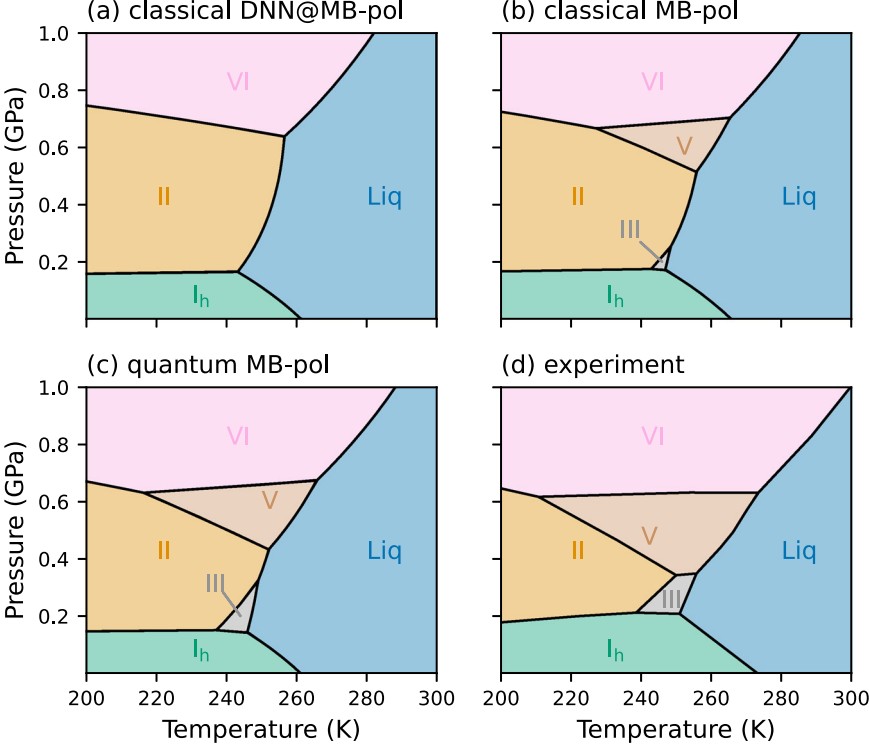

**Fig. 2 | Phase diagram of water.** The phase diagrams calculated at the classical level with DNN@MB-pol (**a**) and MB-pol (**b**), and at the quantum level with MB-pol (**c**) are compared with the experimental phase diagram (**d**). The experimental phase diagram is adapted from ref. 5. The regions of stability for ice $I_h$, II, III, V, and VI and liquid water are shown as areas colored in green, orange, gray, brown, pink, and blue, respectively.

precisely on the estimated coexistence lines. It should be noted that, although the coexistence lines were traced using the DNN@MB-pol potential, the consistency between the melting lines computed using Gibbs-Duhem integration and the melting points determined from enhanced-coexistence simulations is equally good for the corresponding estimates obtained at both classical (Fig. 1b) and quantum (Fig. 1c) levels using the MB-pol potential upon applying thermodynamic perturbation theory and thermodynamic integration by mass, respectively. The comparisons shown in Fig. 1 thus demonstrate that directly tracing the melting lines using Gibbs-Duhem integration from single melting points calculated for each ice polymorph is indeed a reliable approximation.

**Phase diagram**

Starting from the triple points of the liquid-ice coexistence lines, we performed additional Gibbs-Duhem integration calculations to obtain the triple points reported in Supplementary Tables 2–7, and then determine the DNN@MB-pol and MB-pol phase diagrams shown in Fig. 2. The phase diagram calculated at the classical level with the DNN@MB-pol potential (Fig. 2a) correctly locates the regions of stability of ice $I_h$, ice II, and ice VI, but does not predict any region of stability for ice III and ice V. In contrast, the MB-pol phase diagram (Fig. 2b) obtained at the classical level from thermodynamic perturbation of the corresponding DNN@MB-pol phase diagram displays distinct regions of stability for all ice polymorphs, achieving semi-quantitative agreement with the experimental phase diagram (Fig. 2c). Accounting for nuclear quantum effects further expands the regions of stability associated with ice III and ice V, effectively bringing the MB-pol phase diagram calculated at the quantum level to a quantitative agreement with the experimental phase diagram. This trend is consistent with previous observations derived from simulations carried out with different water models[32,35], which highlighted the importance

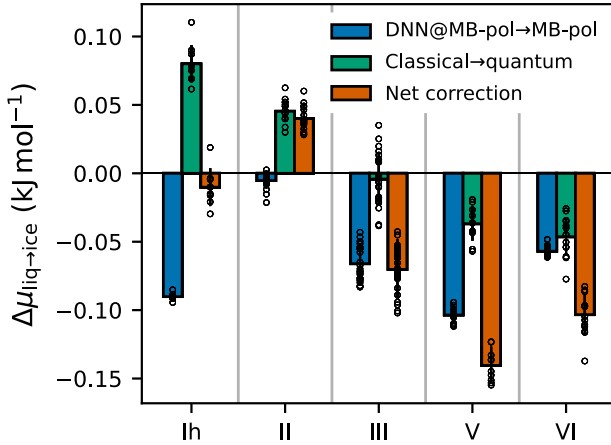

**Fig. 3 | Corrections to chemical potentials.** Average corrections calculated from thermodynamic perturbation (i.e., classical DNN@MB-pol → classical MB-pol) and thermodynamic integration by mass (i.e., classical MB-pol → quantum MB-pol), and corresponding net corrections applied to the differences in chemical potentials between the liquid phase and each ice polymorph. Each error bar is defined as the variance of the correction over the thermodynamic conditions used in the calculations.

of nuclear quantum effects for a correct representation of free-energy differences involving ice III and ice V.

The systematic improvement in the description of the phase boundaries observed when moving from classical DNN@MB-pol to classical MB-pol simulations, and then from classical MB-pol to quantum MB-pol simulations provides fundamental insights into the level of accuracy necessary for achieving a realistic representation of the phase behavior of water. In this regard, Fig. 3 reports the corrections applied to the chemical potential calculated with the DNN@MB-pol potential for each ice polymorph with respect to the liquid phase, which were necessary to elevate the DNN@MB-pol results to the actual MB-pol values (i.e., classical DNN@MB-pol → classical MB-pol, and classical MB-pol → quantum MB-pol). While DNN@MB-pol demonstrates remarkable consistency with MB-pol for energies and forces calculated for molecular configurations extracted from MB-pol simulations carried out over a wide range of thermodynamic conditions (see Supplementary Note 1), the DNN@MB-pol phase diagram calculated at the classical level (Fig. 2a) is qualitatively different from the corresponding MB-pol phase diagram (Fig. 2b). Figure 3 shows that the corrections associated with thermodynamic perturbation calculations that connect DNN@MB-pol to MB-pol (blue bars) overall favor the ice polymorphs over the liquid phase. This leads to a significant contraction of the region of stability for the liquid phase and, consequently, provides space for ice III and ice V, improving the agreement with the experimental phase diagram. The shifts in the stability of the different ice polymorphs are due to the high sensitivity of the free-energy landscape of water to the level of accuracy achieved in the description of the underlying molecular interactions, which emphasizes that the DNN@MB-pol potential, by construction, is not an exact clone of the MB-pol potential. In this regard, it has recently been shown that neural network potentials, such as DNN@MB-pol, are intrinsically limited in their transferability across different phases and thermodynamic conditions, being unable to correctly represent individual many-body contributions to the underlying energy landscape[114], which is particularly important in determining the relative stability of different ice phases[106,115].

Our results are in line with previous observations of proton-ordered ice II being destabilized relative to the other proton-disordered ice phases when long-range interactions are properly accounted for[35]. Based on the results obtained with the MB-pol

potential, we hypothesize that the phase diagram of water reported in ref. 36, which predicts ice III to be thermodynamically unstable, may benefit from thermodynamic perturbation calculations connecting the neural network potential to the actual reference DFT model. It should, however, be noted that the original phase diagram of ref. 36, which was calculated using the Einstein Molecule method, may also change significantly when the free-energy differences are calculated using direct-coexistence or enhanced-coexistence simulations as discussed in Supplementary Note 5.

As shown in Fig. 2, accounting for nuclear quantum effects leads to a quantitative agreement between the MB-pol and experimental phase diagrams. Figure 3 indicates that the corrections to the differences in chemical potential calculated at the quantum-mechanical level are positive for the liquid-ice $I_h$ and liquid-ice II equilibria, negligible for the liquid-ice III equilibrium, and negative for the liquid-ice V and liquid-ice VI equilibria, in line with previous observations based on simulations carried out with the pairwise-additive TIP4PQ/2005[32] model and the revPBE0-D3 DFT model[35]. The different magnitude of the quantum corrections likely depends on the delicate interplay between competing nuclear quantum effects[116] and different hydrogen-bonding topologies of different ice phases[117], which are further modulated by temperature and pressure. The investigation of the different impact that nuclear quantum effects have on the difference in chemical potential between liquid water and different ice polymorphs will be the focus of a future study. Interestingly, the largest shifts between the coexistence lines calculated at the classical and quantum levels with MB-pol are on the order of ~5 K, which are significantly smaller than the shift of ~20 K obtained from simulations with the TIP4PQ/2005 model. These differences can possibly be attributed to the competition between inter- and intra-molecular nuclear quantum effects[116], which is correctly represented by realistic water models such as MB-pol[102] but exaggerated by empirical pairwise additive models such as TIP4PQ/2005[118].

**Comparison with current state-of-the-art simulations**

Figure 4 compares the MB-pol phase diagram calculated at the quantum-mechanical level with current state-of-the-art phase diagrams reported in the literature for various water models. Empirical pairwse-additive water models belonging to the TIP4P family (such as TIP4P/2005[33] and TIP4P/Ice[113] shown in Fig. 4a and b, respectively) and the polarizable iAMOEBA model[34] (Fig. 4c) are able to qualitatively capture some features of the experimental phase diagram. However, none of the regions of stability for the different ice polymorphs is correctly represented, except that for ice $I_h$. In particular, both TIP4P/2005 and TIP4P/Ice largely overestimate the region of stability of ice III, which consequently leads to the shrinking of the region of stability for ice VI and pushes the region of stability for ice II down to temperatures below 100–150 K[33,113]. Similar performance is exhibited by iAMOEBA that places the region of stability for ice II and ice VI at significantly lower temperatures (below 200 K) and higher pressures (above 1.2 GPa), respectively, compared to the experimental phase diagram.

Both DFT-based phase diagrams calculated at the classical level with the DNN@SCAN potential (Fig. 4d) and at the quantum level with the revPBE0-D3 model (Fig. 4e) predict ice $I_h$, ice II, ice V, and ice VI to be stable. However, the predicted regions of stability are significantly different from those observed experimentally, with the revPBE0-D3 model predicting ice VI to be only stable above 1 GPa and below 250 K. In addition, both DNN@SCAN and revPBE0-D3 do not predict ice III to be a stable phase, which is in clear disagreement with the experimental observations. As discussed in Supplementary Note 5, the absence of a region of stability for ice III in the DNN@SCAN and revPBE0-D3 phase diagrams is likely an artifact of the Einstein Molecule method[28] and the closely related Debye Crystal method[119], respectively. While calculating

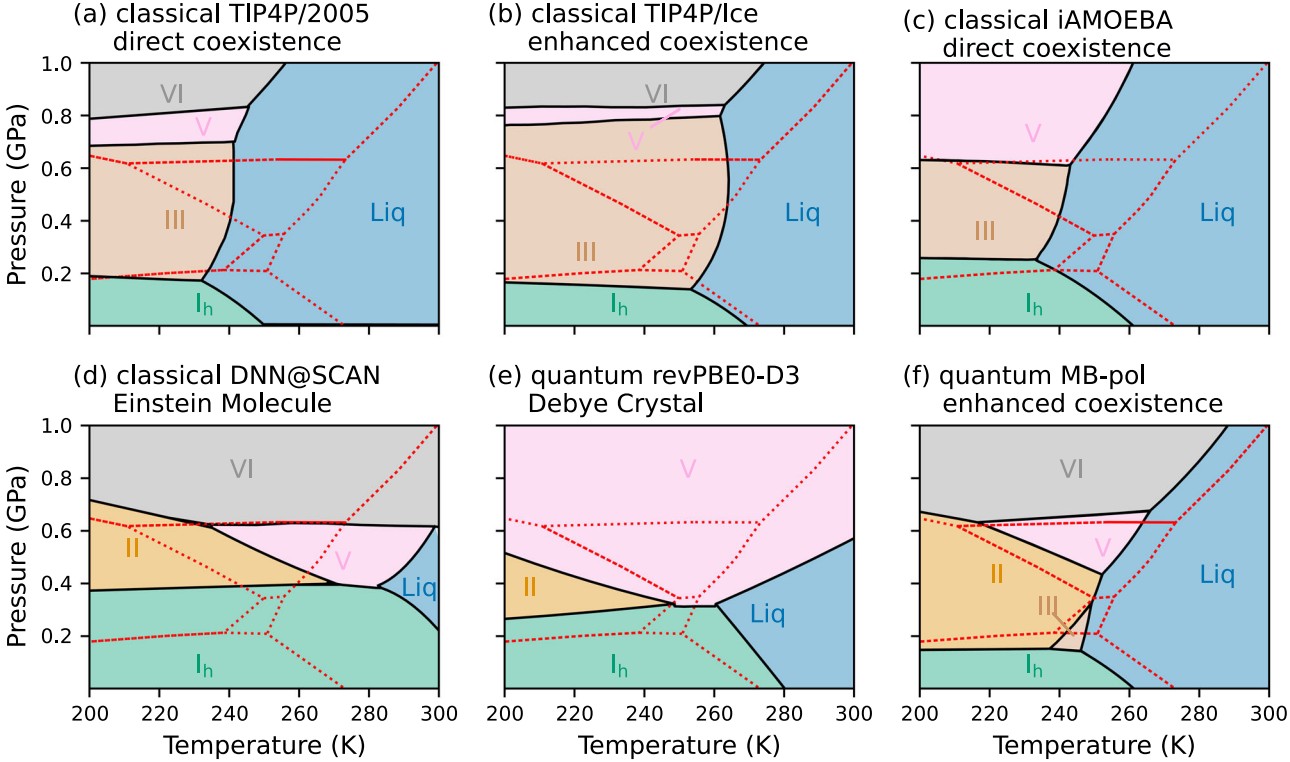

**Fig. 4 | Comparison among state-of-the-art simulations. a** Classical phase diagram of TIP4P/2005 from ref. 33 calculated using direct-coexistence simulations. **b** Classical phase diagram of TIP4P/Ice from ref. 113 calculated using enhanced-coexistence simulations. **c** Classical phase diagram of iAMOEBA from ref. 34 calculated using direct-coexistence simulations. **d** Classical phase diagram of DNN@SCAN from ref. 36 calculated using the Einstein Molecule method. **e** Quantum phase diagram of revPBE0-D3 from ref. 35 calculated using the Debye Crystal method. **f** Quantum phase diagram of MB-pol calculated in this study using enhanced-coexistence simulations. The phase diagrams that are shown in **a**, **c**, **d**, and **e** were digitized from the original references. In each panel, the regions of stability for ice $I_h$, II, III, V, and VI and liquid water are shown as areas colored in green, orange, gray, brown, pink, and blue, respectively, and the experimental phase diagram[5] is shown using dotted red lines.

the DNN@SCAN and revPBE0-D3 phase diagrams using direct-coexistence or enhanced-coexistence simulations will likely result in the appearance of a region of stability for ice III, this will also be accompanied by the shrinking of the region of stability for ice II, which is already underestimated by both water models. In addition, since the region of stability for ice VI is independent of the method used to calculate the phase diagram (Supplementary Fig. 30), the revPBE0-D3 phase diagram calculated using direct-coexistence or enhanced-coexistence simulations will likely still not be able to predict the correct region of stability for ice VI. Figure 4 clearly demonstrates that MB-pol outperforms all models that have, to date, been used to simulate the low-pressure region of the phase diagram of water. Combined with previous findings[97–109], the comparisons shown in Fig. 4 provide further support to the notion that MB-pol currently provides the most realistic representation of water across different phases and thermodynamic conditions.

**Thermodynamic transferability**
While demonstrating remarkable accuracy in predicting the properties of water across the entire phase diagram, MB-pol is still a computer model and, therefore, by definition, does not exactly correspond to "real" water. For example, due to a nearly constant shift of ~10 K in the liquid–ice $I_h$ coexistence line, MB-pol slightly underestimates the melting points of the ice polymorphs. As a result, the melting points predicted by TIP4P/Ice and revPBE0-D3 for ice $I_h$ at 1 atm appear to be in closer agreement with the experimental value (Table 1). Among all water models, MB-pol, however, clearly displays better transferability across different phases and is the only model that correctly reproduce the overall shape of the experimental phase

diagram. Importantly, the deviations between the MB-pol and experimental coexistence lines are always on the order of ~10 K (~0.02 kcal/mol), demonstrating that MB-pol consistently predicts the properties of "real" water across different phases and thermodynamic condition with an accuracy that is well within chemical accuracy (1 kcal/mol)[120].

Accounting for nuclear quantum effects lowers the melting point predicted by MB-pol for ice $I_h$ at 1 atm by 3.9 K. While, on the absolute temperature scale, this shift results in slightly worse agreement with the experimental value, the relative difference between the classical and quantum melting points of ice $I_h$ at 1 atm

**Table 1 | Melting point ($T_m$) and heat of fusion of ice $I_h$ ($H_{fus}$) at 1 atm**

| Method | $T_m$/K | $H_{fus}$/kJ mol$^{-1}$ |
|---|---|---|
| $H_2O$, experiment | 273.15 | 6.01 |
| $D_2O$, experiment | 276.95 | 6.22 |
| TIP4P/Ice | 269.8 | 5.39 |
| DNN@SCAN | 312 | 7.6 |
| $H_2O$, revPBE0-D3 | 276 | 6.8 |
| $D_2O$, revPBE0-D3 | 282 | 7.4 |
| $H_2O$, MB-pol | 262.3 | 5.83 |
| Classical, MB-pol | 266.2 | 6.42 |

Comparisons between the melting points and the heats of fusion determined from computer simulations with TIP4P/Ice[122,137], the DNN@SCAN potential[121], the revPBE0-D3 model[75], and the MB-pol PEF. We estimated the revPBE0-D3 values from the updated chemical potentials that correct a sign error in the original calculations[75] as described in ref. 35.

predicted by MB-pol is in remarkable agreement with the difference of 3.8 K between the melting points of D$_2$O and H$_2$O ice I$_h$ measured experimentally at 1 atm. This agreement is consistent with the notion that classical simulations more closely describe the behavior of heavy water[40]. Moreover, the MB-pol classical and quantum heats of fusion determined at 1 atm from the corresponding chemical potentials (Table 1) are within 3% of the experimental values measured for H$_2$O and D$_2$O ice Ih, respectively. To put the MB-pol results in context, DNN@SCAN overestimates the heats of fusion of H$_2$O and D$_2$O ice I$_h$ by 27% and 22%, respectively[121], revPBE0-D3 overestimates the heats of fusion of H$_2$O and D$_2$O ice I$_h$ by 13% and 18%, respectively[75], and TIP4P/Ice underestimates the heats of fusion of H$_2$O and D$_2$O ice I$_h$ by 10% and 15%, respectively[122].

## Discussion

We have demonstrated that the "first pinciples" MB-pol data-driven many-body potential, which was rigorously derived from the many-body expansion of the energy calculated at the "gold standard" coupled-cluster level of theory, predicts the low-pressure region of the phase diagram of water in quantitative agreement with experiment, exhibiting an unprecedented level of realism for molecular-level computer simulations. Besides marking an important milestone in computer simulations of water, both accuracy and transferability demonstrated by MB-pol across a wide range of thermodynamic conditions provide support for the reliability and validity of MB-pol simulations of water under conditions that are difficult to access by experiments[19,123]. In this context, the close resemblance of MB-pol to the long-sought-after "universal model" of water, as defined in ref. 124, provides a more realistic basis for "in silico" studies of supercooled water and ice nucleation, which have long puzzled the scientific community due to inconsistent or inconclusive results from existing water models. For example, computer simulations with various water models have yielded a wide range of predictions for a possible liquid-liquid critical point[125] and rates of homogeneous ice nucleation[122]. Importantly, since the data-driven many-body formalism originally adopted in the development of the MB-pol potential has recently been extended to generic molecules[126–130], our results also indicate that it will soon be possible to perform realistic molecular simulations of complex systems, thus bridging the gap between computer modeling and experiments.

## Methods

The phase diagram of water was calculated with MB-pol using a multi-stage approach as described in detail in the Supplementary Information. Briefly, we first developed a deep neural network potential (DNN@MB-pol) trained on MB-pol data which enables MB-pol-level molecular dynamics simulations at a fraction of the computational cost associated with actual MB-pol simulations. This speedup was critical to enabling extensive enhanced-coexistence simulations of the relevant liquid-ice polymorph equilibria, which would have been otherwise unaffordable with MB-pol. Second, we calculated the DNN@MB-pol melting points for the relevant ice polymorphs using enhanced-coexistence simulations which, as discussed in the Introduction, are not affected by possible artifacts arising from approximate definitions of proton disorder[33,112,113]. Third, starting from the DNN@MB-pol melting points, we determined the corresponding MB-pol melting points using thermodynamic perturbation theory (Supplementary Note 2). Fourth, we used thermodynamic integration by mass to account for nuclear quantum effects in the liquid-ice polymorph equilibria and thus calculate the MB-pol quantum melting points for the different ice polymorphs. Finally, we used Gibbs–Duhem integration to trace the coexistence lines connecting all melting points and determine the phase diagrams of water (Fig. 2) at the classical level with DNN@MB-pol and MB-pol, and at the quantum level with MB-pol.

## Data availability

Example files used for the enhanced-coexistence simulations are available at PLUMED-NEST (https://www.plumed-nest.org), as plumID:23.001. Input and data files for all simulations, as well as the DNN@MB-pol potential along with the corresponding training set, are available on Zenodo[131]. Any other data generated and analyzed for this study are available from the authors upon request.

## Code availability

All analysis files are available at the Paesani Lab data Repository: https://github.com/paesanilab/Data_Repository. MB-pol reference energies and forces were computed using the MBX software[132], which is available at: https://github.com/paesanilab/MBX. The DNN@MB-pol potential was trained using DeePMD-kit[133], which is available at: https://github.com/deepmodeling/deepmd-kit. All classical molecular dynamics (MD) simulations were carried out with LAMMPS[134] patched with PLUMED[135] and DeePMD-kit[133]. All path-integral molecular dynamics (PIMD) simulations were carried out with i-PI[136], which is available at: https://github.com/i-pi/i-pi. Any additional codes not listed here are available from the authors upon request.

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

## Acknowledgements

We are grateful to Pablo Piaggi for his aid in setting up the environments, including both oxygen and hydrogen atoms, for the enhanced-coexistence simulations involving ice II. We thank Bingqing Cheng for providing the updated figure of the revPBE0-D3 chemical potentials and making the i-PI input files for the revPBE0-D3 simulations available on GitHub, which were of great help in calculating the quantum corrections to the chemical potentials, and Carlos Vega and Venkat Kapil for their comments on the initial version of our manuscript and insights about the Einstein Molecule and Debye Crystal methods. This research was supported by the Air Force Office of Scientific Research grant no. FA9550-20-1-0351 (S.L.B. and F.P.). Computational resources were provided by the Department of Defense High Performance Computing Modernization Program (HPCMP), the Extreme Science and Engineering Discovery Environment (XSEDE), which is supported by the National Science Foundation through grant no. 1548562, the Advanced Cyberinfrastructure Coordination Ecosystem: Services & Support (ACCESS) program, which is supported by National Science Foundation grants nos. 2138259, 2138286, 2138307, 2137603, and 2138296, the Triton Shared Computing Cluster (TSCC) at the San Diego Supercomputer Center (SDSC), and the Scientific Computing Core at the Flatiron Institute, a division of the Simons Foundation.

## Author contributions

F.P. conceived the research. S.L.B. and F.P. designed the research. S.L.B. performed the simulations. S.L.B and F.P. analyzed and discussed the results, and wrote the paper. F.P. acquired funding and administered the project.

## Competing interests

The authors declare no competing interests.
