## [Peer Review File · Nature Communications]

Realistic Phase Diagram of Water from "First Principles" Data-Driven Quantum SimulationsReviewers' comments:

Reviewer #1 (Remarks to the Author):

The manuscript presents a computation of water's phase diagram in the low-pressure regime, which consists of the liquid phase and five ice phases. Theoretical computation of water's phase diagram and determining the phase boundaries accurately are extremely challenging due to the coexistence of complex water-water interactions, entropic effects, and nuclear quantum effects. This work is based on the MB-pol water model, which is mainly developed in the authors' group and is one of the accurate water models built from ab initio at coupled cluster level. It has already been proven in previous works that MB-pol achieves better performance than standard density functional theory models for many properties of water. In this work, the authors study the phase diagram of water and show MB-pol based calculations can reproduce the phase diagram of water/ice in the studied regime, which has been a challenging problem in the community. The results show a significant improvement over that of some widely used classical force field models and DFT models. Technically, this study employs a deep neural network machine learning model trained upon MB-pol, which was combined with enhanced sampling to determine the phase boundaries, then the corrections from DNN@MB-pol to MB-pol and the nuclear quantum effects are included with further simulations. This computational framework is reasonable, and the conclusions are validly supported. The work demonstrates another convincing evidence that complex matters such as water can be understood at the molecular level from ab initio with continuous methodological developments. Overall, the work and the manuscript are of very high quality, and I recommend it is suitable to be published at Nature Communications after minor revisions.

The following are a few specific comments for the authors to consider addressing or clarifying.

(1) I think the title "quantum phase diagram of water" has the risk of confusing in the physics community where "quantum phase diagram" could lead to a link to "quantum phase transition".

However, the phase transitions discussed in this work are all classical phase transitions. Even when the nuclear quantum effects are included, the water-ice and ice-ice phase transitions should still be described by classical statistical mechanics. In addition, the title is also a bit vague and too broad in my opinion. Therefore, I suggest a change of the title to something more precise.

(2) The authors comment, "...that short-range neural network potentials, such as DNN@MB-pol, are intrinsically limited in their ability to represent the long-range components of molecular interactions...".

It might be useful to try out the neural network models with special treatments of long-range potentials, for example, the DPLR model that is readily available in the DEEPM code used by the authors. This could clarify whether the difference in DNN@MB-pol and MB-pol is indeed caused by long-range effects or other model/training details. As the community aims at developing better water models, such a test could provide important information.

(3) Although the current results are already impressively good, there is still a small difference to the experimental benchmark. Is there any chance that the authors could provide some insights on what is still missing in the current model to exactly reproduce the experimental curve, namely an outlook for future work? What is the main factor to account for the remaining errors? The thermodynamic simulations? the model limitation of MB-pol? the accuracy limitation of coupled cluster data used as the training data?

(4) The authors have emphasized "chemical accuracy" of the MB-pol model, but a "chemical accuracy" is on the order of KJ/mol, which is ~ 100 K in terms of temperature. As the authors mentioned "the phase diagram of water require highly precise determination of the associated free-energy landscape since some ice polymorphs are separated energetically by only 0.06 kJ/mol". In ref. 105, the authors have presented MB-pol results on ice polymorphs, and it does not seem to support that MB-pol can reach <0.1 KJ/mol accuracy for the relative energy of different ice polymorphs? A related question would be can coupled cluster reach this level of accuracy? If this accuracy is not reached for the energetics of ice polymorphs, why the phase boundaries in this work can reach the accuracy of a few K. Can the authors comment on this?

(5) The sentence "MB-pol effectively opens the door to realistic in silico studies of homogeneous nucleation, water under confinement, and deeply supercooled water, which have puzzled the scientific

community for decades" need references. In some studies of these related systems, machine learned water models have also been employed.

(6) In the Introduction and Method, when mentioning the enhanced coexistence simulation, both places mentioned the enhanced coexistence simulations do not rely on any approximations for the entropic contribution, which I was not very clear when reading. A brief explanation could be helpful.

Reviewer #2 (Remarks to the Author):

This manuscript provides detailed free energy calculations of water's phase diagram at low pressures using a force-field obtained via machine learning based on forces and energies computed from first-principles calculations. From a technical perspective, the employed methodology is sound and rigorous and is in line with best practices in the field. Moreover, the corresponding author has a proven track record of developing accurate ML-based force-fields for water. However, I find the work not of broad interest to the readership of Nature Communications. It is true that using the quantum-corrected force-field improves the agreement with experiments overall, but the agreement is not perfect. The shifts in phase boundaries are overall smaller than other models, but not by a lot. Moreover, the more accurate—and more expensive—force field employed by the authors does a slightly worse job in predicting water's melting point at ambient pressure than the classical TIP4P/Ice potential. As such, while I find this work an important contribution to the field of computational studies of water, I do not find it significant enough of a contribution to warrant its publication in Nature Communications. Rather, I find it more suitable for specialized journals such as the Journal of Chemical Physics, Journal of Physical Chemistry or Journal of Chemical Theory and Computation.

Reviewer #3 (Remarks to the Author):

The authors present a computational study of the quantum phase diagram of water. They use MB-pol based methods. Their results appear to show accurate phase behavior of water with pressures ranging from 0 to 1 GPa. While the results are interesting, the novelty of this work seems not meeting the high standard of Nature Communications. I suggest the authors address the following issues prior to submitting this manuscript to a more specialized journal such as J. Phys. Chem. Lett.

The authors proposed three steps to compute the quantum phase diagram: Use pre-trained DNN@BM-pol to compute the difference in chemical potential between liquid water and ice phases with enhanced-coexistence simulations; perform the thermodynamic perturbation calculations to estimate the difference between DNN@MB-pol and original MB-pol; consider quantum corrections to the MB-pol classical chemical potentials. The authors should comment on the necessity of DNN@MB-pol. I note that authors' previous study (<https://doi.org/10.1063/5.0142843>) showed the inability of the DNN@MB-pol potential to correctly represent many-body interactions. Why not use MB-pol directly to compute the phase diagram as MB-pol is already affordable for studying phase diagrams, whereas DNN@MB-pol is developed to clone MB-pol?

The authors should comment on why the quantum correction is positive for ice Ih/II and negative for ice III/V/VI in Fig. 3 to bring more physical insight into the manuscript.

In Figure 4. The authors compare their phase diagram with others but the comparison is not comprehensive. The authors stated that enhanced coexistence is better than direct coexistence and the Einstein Molecule method. The results of enhanced-coexistence simulations using other potentials (SCAN/revPBE0+D3) are unavailable for comparison.

Response to Reviewer 1

The manuscript presents a computation of water’s phase diagram in the low-pressure regime, which consists of the liquid phase and five ice phases. Theoretical computation of water’s phase diagram and determining the phase boundaries accurately are extremely challenging due to the coexistence of complex water-water interactions, entropic effects, and nuclear quantum effects. This work is based on the MB-pol water model, which is mainly developed in the authors’ group and is one of the accurate water models built from ab initio at coupled cluster level. It has already been proven in previous works that MB-pol achieves better performance than standard density functional theory models for many properties of water. In this work, the authors study the phase diagram of water and show MB-pol based calculations can reproduce the phase diagram of water/ice in the studied regime, which has been a challenging problem in the community. The results show a significant improvement over that of some widely used classical force field models and DFT models. Technically, this study employs a deep neural network machine learning model trained upon MB-pol, which was combined with enhanced sampling to determine the phase boundaries, then the corrections from DNN@MB-pol to MB-pol and the nuclear quantum effects are included with further simulations. This computational framework is reasonable, and the conclusions are validly supported. The work demonstrates another convincing evidence that complex matters such as water can be understood at the molecular level from ab initio with continuous methodological developments. Overall, the work and the manuscript are of very high quality, and I recommend it is suitable to be published at Nature Communications after minor revisions.

We thank the Reviewer for a detailed and positive assessment of our manuscript, and support for publication.

The following are a few specific comments for the authors to consider addressing or clarifying.

(1) I think the title “quantum phase diagram of water” has the risk of confusing in the physics community where “quantum phase diagram” could lead to a link to “quantum phase transition”. However, the phase transitions discussed in this work are all classical phase transitions. Even when the nuclear quantum effects are included, the water-ice and ice-ice phase transitions should still be described by classical statistical mechanics. In addition, the title is also a bit vague and too broad in my opinion. Therefore, I suggest a change of the title to something more precise.

We thank the Reviewer for the suggestion. We modified the title to “Realistic Phase Diagram of Water from “First Principles” Data-Driven Quantum Simulations” to better capture the essential aspects and ultimate goal of our study.

(2) The authors comment, “. . . that short-range neural network potentials, such as DNN@MB-pol, are intrinsically limited in their ability to represent the long-range components of molecular interactions. . .”. It might be useful to try out the neural network models with special treatments of long-range potentials, for example, the DPLR model that is readily available in the DEEPMD code used by the authors. This could clarify whether the difference in DNN@MB-pol and MB-pol is indeed caused by long-range effects or other model/training details. As the community aims at developing better water models, such a test could provide important information.

This is a very interesting point. In parallel with the present study, we carried out an extensive and thorough study of the performance of various DeePMD potentials derived from different training sets, with and without long-range contributions. Our paper has recently appeared in the Journal of Chemical Physics (<https://doi.org/10.1063/5.0142843>). In this study, we demonstrate that the current DeePMD architecture is unable to correctly represent individual many-body effects, which largely limits the transferability of DeePMD potentials across different phases and thermodynamic conditions. As shown in Figures S4 and S5 of the Supplementary Material for our J. Chem. Phys. paper (<https://doi.org/10.1063/5.0142843>), the explicit inclusion of long-range contributions does not improve the ability of DeePMD potentials to correctly reproduce individual many-body contributions.

We agree with the Reviewer that it is very important to assess the reliability of neural network potentials of water, especially in tests that allow for determining their ability to generalize. In this context, we would like to point out that the analyses reported in our J. Chem. Phys. paper (<https://doi.org/10.1063/5.0142843>) show that equivariant graph neural network potentials are also unable to correctly represent many-body interactions in water. These findings seem to suggest (at least to us) that the inability to correctly represent many-body interactions is likely common to all neural network potentials that do not explicitly encode individual many-body contributions into their architectures, independently of whether long-range interactions are included or not.

We modified the text in order to be more specific about the conclusions derived from our J. Chem. Phys. study.

(3) Although the current results are already impressively good, there is still a small difference to the experimental benchmark. Is there any chance that the authors could provide some insights on what is still missing in the current model to exactly reproduce the experimental curve, namely an outlook for future work? What is the main factor to account for the remaining errors? The thermodynamic simulations? the model limitation of MB-pol? the accuracy limitation of coupled cluster data used as the training data?

These are very interesting questions that we have also been asking ourselves since we introduced MB-pol in 2013-2014. On the one hand, we find it somewhat reassuring that MB-pol is not “exact” since, while being highly accurate, it is still a computer model. On the other hand, we think that there may still be room for improvement. For example, the current version of MB-pol is not explicitly dissociable. This implies that, while the energies of highly distorted monomers are still accurately represented by the short-range 2-body and 3-body permutationally invariant polynomials used by MB-pol, the more “ionic” contributions to the hydrogen bonds may still be underestimated.

From a more technical perspective, as mentioned in *Acc. Chem. Res.* 49, 1844 (2016), MB-pol can systematically be improved in various ways, for example: 1) by training the 2-body and 3-body permutationally invariant polynomials on larger training sets of water dimers and trimers, 2) by adding permutationally invariant polynomials for higher n -body terms (i.e., 4-body, 5-body, etc.) explicitly trained on CCSD(T) reference data calculated in the complete basis set limit, and 3) by adopting higher-order and more expensive permutationally invariant polynomials in the representation of n -body energies. Work along these lines is ongoing in our group.

The differences between the MB-pol and experimental phase diagrams may also be due to finite size effects in the enhanced-coexistence simulations as well as to approximating the quantum partition function with Feynman’s ring-polymers consisting of 32 beads. We believe that, as both hardware and software continue to improve, path-integral molecular dynamics simulations using MB-pol for larger systems and number of beads will become possible in the near future.

(4) The authors have emphasized “chemical accuracy” of the MB-pol model, but a “chemical accuracy” is on the order of KJ/mol, which is 100 K in terms of temperature. As the authors mentioned “the phase diagram of water require highly precise determination of the associated free-energy landscape since some ice polymorphs are separated energetically by only 0.06 kJ/mol”. In ref. 105, the authors have presented MB-pol results on ice polymorphs, and it does not seem to support that MB-pol can reach <0.1 KJ/mol accuracy for the relative energy of different ice polymorphs? A related question would be can coupled cluster reach this level of accuracy? If this accuracy is not reached for the energetics of ice polymorphs, why the phase boundaries in this work can reach the accuracy of a few K. Can the authors comment on this?

In our study, we use the standard definition of chemical accuracy as an error of 1 kcal/mol indirectly introduced by Pople in his Noble Lecture [*Rev. Mod. Phys.* 71, 1267 (1999)], and then adopted in the computational chemistry literature [e.g., see *J. Chem. Phys.* 116, 1493 (2002); *Phys. Chem. Chem. Phys.* 8, 4398 (2006); *J. Chem. Phys.* 126, 114105 (2007); *Proc. Natl. Acad. Sci. U.S.A.* 108, 19896 (2011); *Nat. Commun.* 11, 5223 (2020)]. Based on this definition, MB-pol does not only achieve chemical accuracy, but actually exhibits sub-chemical accuracy for most properties of water, from the gas to the condensed phase [e.g., see *Acc. Chem. Res.* 49, 1844 (2016) and *J. Chem. Phys.* 145, 194504 (2016)]. This is the reason why MB-pol is the first, and currently, only water model able to accurately reproduce the vibration-rotation tunneling spectrum of the water dimer [*J. Chem. Theory Comput.* 9, 5395 (2013)], the quantum equilibria in water clusters [*Science* 351, 1310 (2016); *Science* 352, 1194 (2016); *J. Am. Chem. Soc.* 139, 7082 (2017)], the structural and thermodynamic properties of liquid water (*J. Chem. Phys.* 145, 194504 (2016)], the infrared and Raman spectra of liquid water (*J. Chem. Theory Comput.* 11, 1145 (2015); *J. Chem. Phys.* 147, 244504 (2017)], the vapor–liquid equilibrium properties (*J. Chem. Phys.* 154, 211103 (2021)); the sum-frequency generation spectra of the water surface (*J. Am. Chem. Soc.* 138, 3912 (2016); *J. Phys. Chem. B* 122, 4356 (2018)], the infrared and Raman spectra of various ice phases (*J. Phys. Chem. Lett.* 8, 2579 (2017); *J. Phys. Chem. B* 122, 10572 (2018); *Proc. Natl. Acad. Sci. U.S.A.* 116, 24413 (2019), the structural and thermodynamic properties of supercooled water (*J. Phys. Chem. Lett.* 13, 3652 (2022)], and now the phase diagram of water.

The ability of CCSD(T) calculations to achieve “chemical accuracy” in the representation of molecular interactions is well established in the literature (e.g., see Refs. 51 and 52 of the revised manuscript).

(5) The sentence “MB-pol effectively opens the door to realistic in silico studies of homogeneous nucleation, water under confinement, and deeply supercooled water, which have puzzled the scientific community for decades” need references. In some studies of these related systems, machine learned water models have also been employed.

While we agree with the Reviewer that machine-learned water models have already been employed to study ice nucleation and supercooled water, our present study clearly demonstrates (Figure 4 of the revised manuscript) that none of these models provides a “realistic” representation of water.

To provide further evidence for the inadequacy of existing force fields and machine-learned models to represent water, the figure below shows comparisons between the experimental values of the isothermal compressibility of water measured as a function of temperature at 1 atm and the corresponding values calculated with various models, including TIP4P/2005 and iAMOEBA (panel A), DNN@SCAN (panel B), and MB-pol (panel C), whose phase diagrams are compared in Figure 4 of our revised manuscript. We believe that it is quite evident that TIP4P/2005, iAMOEBA, and DNN@SCAN are unable to provide a “realistic” representation of supercooled water. In contrast, MB-pol closely reproduces the experimental values that are available from the boiling point down to ~ 230 K. This implies that MB-pol is the only water model that currently provides a realistic representation of water in the so-called “no man’s land” region of the phase diagram (i.e., temperatures below 230 K and higher pressures) that has to date proven difficult to probe experimentally.

Figure R1: Comparisons between the experimental values of the isothermal compressibility of water measured as a function of temperature and the corresponding values calculated with various models.

(6) In the Introduction and Method, when mentioning the enhanced coexistence simulation, both places mentioned the enhanced coexistence simulations do not rely on any approximations for the entropic contribution, which I was not very clear when reading. A brief explanation could be helpful.

Direct-coexistence and enhanced-coexistence simulations explicitly simulate the crystallization process, which implies that they naturally account for entropic contributions associated with proton disorder in calculations of free-energy differences between liquid water and a given ice polymorph, independently of the extent of proton disorder present in the ice polymorph.

A detailed description of both algorithms is reported in Refs. 28, 33, and 113. We modified the text of the revised manuscript to clarify this point and refer the reader to the original references.

Response to Reviewer 2

This manuscript provides detailed free energy calculations of water’s phase diagram at low pressures using a force-field obtained via machine learning based on forces and energies computed from first-principles calculations. From a technical perspective, the employed methodology is sound and rigorous and is in line with best practices in the field. Moreover, the corresponding author has a proven track record of developing accurate ML-based force-fields for water. However, I find the work not of broad interest to the readership of Nature Communications. It is true that using the quantum-corrected force-field improves the agreement with experiments overall, but the agreement is not perfect. The shifts in phase boundaries are overall smaller than other models, but not by a lot. Moreover, the more accurate—and more expensive—force field employed by the authors does a slightly worse job in predicting water’s melting point at ambient pressure than the classical TIP4P/Ice potential. As such, while I find this work an important contribution to the field of computational studies of water, I do not find it a significant enough of a contribution to warrant its publication in Nature Communications. Rather, I find it more suitable for specialized journals such as the Journal of Chemical Physics, Journal of Physical Chemistry or Journal of Chemical Theory and Computation.

We appreciate the time and effort that the Reviewer has taken to read our manuscript and provide their feedback. While we respect the Reviewer’s opinions, we respectfully disagree with the overall assessment of our work. We believe that there is some confusion, which may, in part, be due to how we presented our results in the original Figure 4.

Panel (a) of the original Figure 4 showed the phase diagram calculated with the TIP4P/ice model using the Einstein Molecule method. As discussed in the literature (Refs. 33 and 113 of the revised manuscript), the Einstein Molecule method significantly underestimates the region of stability of ice III. The correct phase diagram for TIP4P/Ice was reported in panel (b) of the original Figure 4, which was calculated from enhanced-coexistence simulations in Ref. 113 of the revised manuscript. The correct phase diagram of TIP4P/Ice shows that the regions of stability for ice III and ice V are largely incorrect, and ice II is completely missing. Although we had included a specific discussion about the intrinsic limitations of the Einstein Molecule method and superior performance of direct-coexistence and enhanced-coexistence simulations in the original manuscript, we have realized that it was not completely straightforward to connect that discussion in the manuscript to the results presented in the original Figure 4.

Since the phase diagrams for DNN@SCAN (panel d) and revPBE0-D3 (panel c) have only been reported in the literature from calculations carried out with the Einstein Molecule method, we had originally decided to show the TIP4P/Ice phase diagrams calculated using both the Einstein Molecule method (panel a) and enhanced-coexistence simulations (panel b) in order to provide the reader with a means to “imagine” how the DNN@SCAN and revPBE0-D3 phase diagrams may change if they are calculated using direct-coexistence and enhanced-coexistence simulations.

To avoid further confusion, we modified Figure 4 by replacing the original panel (a) with the phase diagram of TIP4P/2005 calculated using direct-coexistence simulations in Ref. 33, and moved the comparisons of the phase diagrams calculated using the Einstein Molecule method and direct-coexistence or enhanced-coexistence simulations to the Supplementary Information. The revised Figure 4 (also shown below), clearly demonstrates that none of the state-of-the-art water phase diagrams that have been reported in the literature is even comparable with the phase diagram predicted by MB-pol. In particular, all existing models are unable to reproduce, even qualitatively, the overall shape of the phase diagram, largely underestimating/overestimating the regions of stability of some ice polymorphs (e.g., ice III for TIP4P/2005, TIP4P/Ice, and iAMOEBA, ice VI for DNN@SCAN, and ice V for revPBE0-D3), and misplacing (e.g., ice II for TIP4P/2005, TIP4P/Ice, and iAMOEBA) or completely missing the regions of stability of other ice polymorphs (i.e., ice III for DNN@SCAN and revPBE0-D3). The comparisons shown in the revised Figure 4 provide unambiguous evidence that MB-pol indeed represents a “quantum leap” in computer simulations of water, correctly reproducing all phase boundaries, which clearly differ by *a lot* from those predicted by any of the other water models.

Regarding the shifts in the melting points of ice I_h, we should note that ~ 10 K corresponds to ~ 0.02 kcal/mol, which is well beyond “chemical accuracy” and, possibly, the intrinsic precision of any correlated electronic structure method currently available (see Refs. 51 and 52). The important and key point, however, is that, contrary to any other water model, MB-pol consistently displays the same accuracy for all phases of water over a wide range of temperatures and pressures, demonstrating that it accurately represents energy differences across all phases of water. In this regard, it should be noted that Table 1 indicates that, while the classical and quantum versions of MB-pol predict melting points for ice I_h at 1 atm that are ~ 6 and ~ 10 K below the experimental value, they predict enthalpies of fusion (6.42 kJ mol⁻¹ and 5.83 kJ mol⁻¹, respectively) that are in better agreement with the experimental value (6.01 kJ mol⁻¹) than any of the other water models. These comparisons suggest that, while it sometimes appear that empirical pairwise-additive models apparently give the “right answers”, they are able to do that by relying on fortuitous error compensation as already discussed in the literature [e.g., see J. Chem. Theory Comput. 9, 1103 (2013) and Chem. Rev. 116, 7501 (2016)]. In contrast, MB-pol consistently provides the “right answers for the right reasons” [e.g., see Acc. Chem. Res. 49, 1844 (2016)].

In conclusion, we believe that the revised Figure 4 unambiguously addresses all of the Reviewer’s concerns. At the same time, a critical assessment of the differences in melting points indicates that MB-pol is as good as a data-driven model can be, based on the inherent accuracy/precision of “gold standard” CCSD(T) calculations.

Figure R2: (a) Classical phase diagram of TIP4P/2005 from Ref. 33 calculated using direct-coexistence simulations. (b) Classical phase diagram of TIP4P/Ice from Ref. 113 calculated using enhanced-coexistence simulations. (c) Classical phase diagram of iAMOEBA from Ref. 34 calculated using direct-coexistence simulations. (d) Classical phase diagram of DNN@SCAN from Ref. 36 calculated using the Einstein Molecule method. (e) Quantum phase diagram of revPBE0-D3 from Ref. 35 calculated using the Einstein Molecule method. (f) Quantum phase diagram of MB-pol calculated in this study using enhanced-coexistence simulations. The phase diagrams that are shown in (a), (c), (d), and (e) were digitized from the original references. The experimental phase diagram from Ref. 5 is shown in each panel using a dotted red line.

Response to Reviewer 3

The authors present a computational study of the quantum phase diagram of water. They use MB-pol based methods. Their results appear to show accurate phase behavior of water with pressures ranging from 0 to 1 GPa. While the results are interesting, the novelty of this work seems not meeting the high standard of Nature Communications. I suggest the authors address the following issues prior to submitting this manuscript to a more specialized journal such as J. Phys. Chem. Lett.

We appreciate the time and effort that the Reviewer has taken to read our manuscript and provide their feedback. While we respect the Reviewer’s opinions, we respectfully disagree with the overall assessment of our work. We believe that there is some confusion that may, in part, be due to how we presented our results in Figure 4.

Panel (a) of the original Figure 4 showed the phase diagram calculated with the TIP4P/ice model using the Einstein Molecule method. As discussed in the literature (Refs. 33 and 113 of the revised manuscript), the Einstein Molecule method significantly underestimates the region of stability of ice III. The correct phase diagram for TIP4P/Ice was reported in panel (b) of the original Figure 4, which was calculated from enhanced-coexistence simulations in Ref. 113 of the revised manuscript. The correct phase diagram of TIP4P/Ice shows that the regions of stability for ice III and ice V are largely incorrect, and ice II is completely missing. Although we had included a specific discussion about the intrinsic limitations of the Einstein Molecule method and superior performance of direct-coexistence and enhanced-coexistence simulations in the original manuscript, we have realized that it was not completely straightforward to connect that discussion in the manuscript to the results presented in the original Figure 4.

Since the phase diagrams for DNN@SCAN (panel d) and revPBE0-D3 (panel c) have only been reported in the literature from calculations carried out with the Einstein Molecule method, we had originally decided to show the TIP4P/Ice phase diagrams calculated using both the Einstein Molecule method (panel a) and enhanced-coexistence simulations (panel b) in order to provide the reader with a means to “imagine” how the DNN@SCAN and revPBE0-D3 phase diagrams may change if they are calculated using direct-coexistence and enhanced-coexistence simulations.

To avoid further confusion, we modified Figure 4 by replacing the original panel (a) with the phase diagram of TIP4P/2005 calculated using direct-coexistence simulations in Ref. 33, and moved the comparisons of the phase diagrams calculated using the Einstein Molecule method and direct-coexistence or enhanced-coexistence simulations to the Supplementary Information. The revised Figure 4 (also shown below), clearly demonstrates that none of the state-of-the-art water phase diagrams that have been reported in the literature is even comparable with the phase diagram predicted by MB-pol. In particular, all existing models are unable to reproduce, even qualitatively, the overall shape of the phase diagram, largely underestimating/overestimating the regions of stability of some ice polymorphs (e.g., ice III for TIP4P/2005, TIP4P/Ice, and iAMOEBA, ice VI for DNN@SCAN, and ice V for revPBE0-D3), and misplacing (e.g., ice II for TIP4P/2005, TIP4P/Ice, and iAMOEBA) or completely missing the regions of stability of other ice polymorphs (i.e., ice III for DNN@SCAN and revPBE0-D3). The comparisons shown in the revised Figure 4 provide unambiguous evidence that MB-pol indeed represents a “quantum leap” in computer simulations of water, correctly reproducing all phase boundaries, which clearly differ by *a lot* from those predicted by any of the other water models.

The authors proposed three steps to compute the quantum phase diagram: Use pre-trained DNN@BM-pol to compute the difference in chemical potential between liquid water and ice phases with enhanced-coexistence simulations; perform the thermodynamic perturbation calculations to estimate the difference between DNN@MB-pol and original MB-pol; consider quantum corrections to the MB-pol classical chemical potentials. The authors should comment on the necessity of DNN@MB-pol. I note that authors’ previous study(<https://doi.org/10.1063/5.0142843>) showed the inability of the DNN@MB-pol potential to correctly represent many-body interactions. Why not use MB-pol directly to compute the phase diagram as MB-pol is already affordable for studying phase diagrams, whereas DNN@MB-pol is developed to clone MB-pol?

We believe that there is some confusion in assessing the overall protocol used by us to calculate the phase diagram of water. As discussed in detail in Supplementary Note 1 and Supplementary Note 2, DNN@MB-pol provides a speedup of $\sim 20\times$ relative to analogous MB-pol molecular dynamics simulations carried out with LAMMPS (Ref. 27 of the SI) patched with MBX (Ref. 23 of the SI). This speedup is critical to enabling extensive enhanced-sampling simulations that are required to determine the melting points and, consequently, the phase diagram of water. The same extensive enhanced-sampling simulations with MB-pol are currently *unaffordable* due the high associated computational cost.

The authors should comment on why the quantum correction is positive for ice Ih/II and negative for ice III/V/VI in Fig. 3 to bring more physical insight into the manuscript.

This is an interesting point which, given the complexity of nuclear quantum effects in water (e.g., see Refs. 40 and 41), we believe, deserves a dedicated study. Our current hypothesis is that these differences depend on the delicate interplay between competing nuclear quantum effects in water [J. Chem. Phys. 131, 024501 (2009)] and hydrogen-bonding topologies of different ice phases [Proc. Natl. Acad. Sci. U.S.A. 108, 6369 (2011)], which are further modulated by temperature and pressure. This aspect of our simulations will be the subject of future work.

Figure R3: (a) Classical phase diagram of TIP4P/2005 from Ref. 33 calculated using direct-coexistence simulations. (b) Classical phase diagram of TIP4P/Ice from Ref. 113 calculated using enhanced-coexistence simulations. (c) Classical phase diagram of iAMOEBA from Ref. 34 calculated using direct-coexistence simulations. (d) Classical phase diagram of DNN@SCAN from Ref. 36 calculated using the Einstein Molecule method. (e) Quantum phase diagram of revPBE0-D3 from Ref. 35 calculated using the Einstein Molecule method. (f) Quantum phase diagram of MB-pol calculated in this study using enhanced-coexistence simulations. The phase diagrams that are shown in (a), (c), (d), and (e) were digitized from the original references. The experimental phase diagram from Ref. 5 is shown in each panel using a dotted red line.

We added a sentence to the revised manuscript to clarify this point.

In Figure 4. The authors compare their phase diagram with others but the comparison is not comprehensive. The authors stated that enhanced coexistence is better than direct coexistence and the Einstein Molecule method. The results of enhanced-coexistence simulations using other potentials (SCAN/revPBE0+D3) are unavailable for comparison.

We believe that some clarification is in order. In no place of our manuscript do we state that enhanced-coexistence simulations are superior to direct-coexistence simulations. On the contrary, we demonstrated in Ref. 113 that these two methods consistently predict similar coexistence lines.

What we state instead is that the Einstein Molecule method provides an incorrect phase boundary for ice III. This was demonstrated by the Madrid group, which developed both the Einstein Molecule (Ref. 28) and direct-coexistence methods (Ref. 33), as well as by us, who developed the enhanced-coexistence method (Ref. 113).

Figure 4 shows all phase diagrams calculated with state-of-the-art water models which have been reported in the literature to date. Although we agree with the Reviewer that it would also be nice (and useful) to compare the DNN@SCAN and revPBE0-D3 phase diagrams calculated from direct-coexistence or enhanced-coexistence simulations, such phase diagrams are not available in the literature. Since calculating the phase diagram of water with “first principles” water models is a daunting and expensive task that requires several months of CPU time (this is the main reason why only a handful of these calculations have, to date, been reported in the literature), unfortunately, we are not in the position to fill this gap. We can, however, still make some observations based on our extensive analyses of various DFT models for water. We believe that our observations provide fundamental insights that can be used to address the Reviewer’s comments:

1. In the case of revPBE0-D3, we demonstrated that, contrary to MB-pol, this functional is unable to correctly capture the delicate balance between 2-body and 3-body interactions in water [Chem. Sci. 10, 8211 (2019)], which limits the ability of revPBE0-D3 to accurately predict the properties of water across different phases. These limitations clearly manifest in revPBE0-D3 being unable to even qualitatively predict the correct shape of the phase diagram of water, as seen in panel (e) of Figure 4. In particular, comparisons with the TIP4P/2005 and TIP4P/Ice phase diagrams calculated with different approaches reported in Supplementary Figure 28 indicate that, while calculating the revPBE0-D3 phase diagram using direct-coexistence or enhanced-coexistence simulations will likely result in the appearance of a region of stability for ice III, this will also be accompanied by the shrinking of the region of stability for ice II, which is already largely underestimated by revPBE0-D3. In addition, the revPBE0-D3 phase diagram calculated with the Einstein Molecule method does not predict the correct region of stability for ice VI which, as shown in Supplementary Figure 28, is independent of the method used

to calculate the phase diagram. This implies that the revPBE0-D3 phase diagram calculated using direct-coexistence or enhanced-coexistence simulations will likely still lack the correct region of stability for ice VI.

2. In the case of SCAN, we demonstrated that this functional exhibits very large errors at the 2-body level, which results in significant overbinding [J. Chem. Theory Comput. 17, 3739 (2021)]. This limitation clearly manifests in DNN@SCAN predicting a phase diagram of water that is, overall, appreciably shifted to higher temperatures, as seen in panel (d) of Figure 4. As in the case of revPBE0-D3, it is expected that calculating the DNN@SCAN phase diagram using direct-coexistence or enhanced-coexistence simulations will likely result in the appearance of a region of stability for ice III, with a consequent shrinking of the regions of stability for both ice II and ice V, which will worsen the agreement with the experimental phase diagram for these two ice polymorphs. In addition, the DNN@SCAN phase diagram calculated with the Einstein Molecule method largely overestimates the region of stability for ice VI which, as shown in Supplementary Figure 28, will likely still be overestimated by direct-coexistence or enhanced-coexistence simulations since both size and shape of the region of stability for ice VI are independent of the method used to calculate the phase diagram.

The inability of DNN@SCAN to correctly model the phase of diagram of water also manifests in DNN@SCAN being unable to provide a “realistic” representation of liquid water. This is demonstrated in Figure R5 that shows the temperature dependence of the isothermal compressibility of water calculated with DNN@SCAN (panel A) and MB-pol (panel B). Consistent with the shifted phase diagram, DNN@SCAN predicts a maximum of the isothermal compressibility that is much shallower and shifted to appreciably higher temperatures compared to the experimental value. In contrast, MB-pol closely reproduces the experimental data that are available from the boiling point down to ~ 230 K.

Figure R4: Comparisons between the experimental values of the isothermal compressibility of water measured as a function of temperature and the corresponding values calculated with DNN@SCAN (panel A) and MB-pol (panel B).

3. It is important to mention that recent work by us and others demonstrated that the limitations discussed above for the revPBE0-D3 and SCAN functionals are general and common to all DFT models due to functional-driven and density-driven errors that are intrinsic to DFT (e.g., see Refs. 65-69). This suggests that none of the existing DFT models is able to correctly represent many-body interactions in water across different phases which, in turn, implies that none of the existing DFT models is able to provide a realistic description of the phase diagram of water.

We revised our manuscript to better emphasize the differences between phase diagrams calculated using different methods. We also expanded the discussion in Supplementary Note 5 about the region of stability for ice III as predicted by the Einstein Molecule method. We believe that these modifications, which are effectively summarized in the revised Figure 4, clearly demonstrate that the MB-pol phase diagram reported in our study represents a breakthrough in computer simulations of water, providing definitive evidence for MB-pol being the long-sought-after “universal” model capable of accurately describe the properties of water across all phases (Ref. 125).

REVIEWERS' COMMENTS

Reviewer #1 (Remarks to the Author):

My comments/questions have been addressed/clarified.

Reviewer #2 (Remarks to the Author):

The authors have provided convincing arguments to address some of my concerns regarding why MB-Pol has a larger error than, say, TIP4P/Ice in predicting the experimental melting point of Ice I. Nonetheless, I still think that these are minor technical details. I therefore stand by my original assessment that this work is more suited for a specialized journal, and is not of enough broad interest to warrant publication in Nature Communications.

Response to Reviewer 1

My comments/questions have been addressed/clarified.

We thank the Reviewer for supporting the publication of our manuscript

Response to Reviewer 2

The authors have provided convincing arguments to address some of my concerns regarding why MB-Pol has a larger error than, say, TIP4P/Ice in predicting the experimental melting point of Ice I. Nonetheless, I still think that these are minor technical details. I therefore stand by my original assessment that this work is more suited for a specialized journal, and is not of enough broad interest to warrant publication in Nature Communications.

We thank the Reviewer for a careful reading of our revised manuscript. While we appreciate the Reviewer's feedback, we respectfully disagree with the overall assessment of the MB-pol and TIP4P/Ice results. We believe that the comparisons presented in Figure 4 unambiguously demonstrate the superior performance of MB-pol. In particular, while TIP4P/Ice does reproduce the melting point of ice more precisely as it was specifically parameterized to do so, its predictions of other properties, including the phase diagram analyzed in our study, are largely incorrect. In contrast, MB-pol, which was rigorously and entirely derived from "first principles", accurately predicts all properties of water from small clusters to liquid water and ice and, as shown in Figure 4, is the first and, currently, only water model that correctly reproduces the phase diagram of water over a wide range of temperatures and pressures.